# Smart and Sustainable Cities: The Main Guidelines of City Statute for Increasing the Intelligence of Brazilian Cities

**Evandro Gonzalez Lima** [1], **Christine Kowal Chinelli** [1], **Andre Luis Azevedo Guedes** [1], **Elaine Garrido Vazquez** [2], **Ahmed W. A. Hammad** [3], **Assed Naked Haddad** [4] **and Carlos Alberto Pereira Soares** [1,*]

1   Programa de Pós-Graduação em Engenharia Civil, Universidade Federal Fluminense, Rio de Janeiro 24210-240, Brazil; evandrogl@id.uff.br (E.G.L.); cchinelli@id.uff.br (C.K.C.); andre.gueves@gmail.com (A.L.A.G.)
2   Departamento de Construção Civil, Universidade Federal do Rio de Janeiro, Rio de Janeiro 21941-909, Brazil; elainevazquez@poli.ufrj.br
3   Faculty of Built Environment, University of New South Wales, 24210-240 Sidney, Australia; a.hammad@unsw.edu.au
4   Programa de Engenharia Ambiental, Universidade Federal do Rio de Janeiro, Rio de Janeiro 21941-909, Brazil; assed@poli.ufrj.br
*   Correspondence: capsoares@id.uff.br; Tel.: +55-21-2629-5410

**Abstract:** The regulation of urban property use is a fundamental instrument for the development of cities. However, most of the norms that set general guidelines for urban policy predate the transformations that the smart city concept has brought about in the way cities are appropriated and perceived by society, and even today, studies on how these regulations collaborate to make cities smarter and more sustainable. This work contributes to filling this gap by investigating the main guidelines of the Brazilian City Statute that have the greatest potential to contribute to having smarter and more sustainable Brazilian cities. To prioritize the sixteen guidelines of the City Statute, the methodology used consisted of a survey carried out with professionals working in the concerned field. The results show that the sixteen guidelines were evaluated as important for increasing the intelligence of cities, of which five were evaluated as having the most priority, these five were related to the governance of cities. Considering the scarcity of resources in Brazilian cities, these five guidelines contribute so that municipal governments can direct their efforts towards what has the most priority.

**Keywords:** smart city; sustainable city; city statute; urban policy; urban planning

## 1. Introduction

Regulation of urban property use has been a fundamental instrument for cities to develop considering the collective good, the welfare of citizens and sustainable development. As the ties between continents tightened, researchers devoted more attention to describing and evaluating land-use systems in different countries [1] and to study the similarities and differences between legal systems, the results of which showed patterns of change in the legal regimes of an evolving world [2]. The catastrophic London fire of 1666 transformed society's understanding of why individual property rights should, to some degree, be subject to the greatest public interest when common challenges are faced [3]. Episodes such as this have triggered important discussions about the extent to which private rights and the private sector should be regulated, the government's competence to direct market forces,

the appropriate role of municipal, state and federal agencies in land use and on appropriate legal regulatory techniques that government should employ to protect the public interest [2].

Main international urban legislation that has inspired other countries include: (a) England: Town and Country Planning Act, 1947, recast in 1990; (b) Spain: Law of the Urban Regiment and Ordinance of 1956, reformulated in 1975 and 1992; (c) Italy: Legge Urbanistica, 1942, reformulated in 1967 and 1977; (d) France: Code de l'Urbanisme et de l'Habitation, 1954, reformulated in 1973; (e) Germany: Bundesbaugesetz (Federal Law on Urban Planning) of 1960 [4]. Although each of these laws have particularities arising from each country's institutional system, there is a striking similarity between them. They all establish a hierarchical system of territorial ordering, whereby the smaller scale plans detail the larger ones. Each of these plans are thoroughly described in terms of their content, approval and updating, degree of detail and legal effectiveness [5].

In Latin America, urban planning legislation was strongly influenced by the uneven urban development experienced by municipal and national governments. This form of development was characterized by occupations and land uses formalized by governmental plans and actions coexisting with informal urban conditions produced by a poor, marginalized and dispossessed civil, social, political system and economic rights [6]. From the 1970s onwards, the growing awareness of environmental problems led to the emergence of the concept of sustainable development [7], intensifying the demands for more sustainable cities. Initiatives focused on improving urban services and cities' infrastructure took place around the world [8,9], based on the perception that the balanced relationship between environmental, economic and social conditions leads to sustainable development cities [10]. In this context, sustainable urban planning is an important instrument for the government and citizens to contribute to urban sustainability, by controlling urbanization zones and land uses [11], being a fundamental instrument for the operationalization of land use and occupation laws. Various social movements, under the banner "Right to the City", have promoted discussions and actions aimed at guaranteeing more sustainable, fair and democratic cities, focusing mainly on the policies and actions of municipal governments, since they are responsible for regulation of land-use and occupation, defining patterns of urban density, infrastructure, among other issues directly related to the territorial development of cities [5]. However, it was only from the 1980s that in Latin America these movements, mainly represented by community movements, social activists and urban reform forums, intensified, leading to the creation of new laws [12] that limited private property rights from the social function of property [13], consolidating the understanding of protection of the right to property subordinated to social or collective interests [14]. They also consolidated the rights to decent housing and a more sustainable environment [13] and incorporated ecological functions into the social function of property [15].

Land use and occupation laws vary in aspiration, ambition and complexity due to cultural, historical, political and geographical differences, and have enabled local governments to act or show awareness of their critical role in achieving sustainable development by addressing the emerging problems of each society, providing appropriate strategies for the culture and place of its origin and involving citizens in the planning and formulation of these policies [16,17]. Thus, land use planning and control aims to organize and control the pattern of urban occupation and expansion in order to ensure that social functions develop in harmony with the urban fabric and that city development occurs in a balanced and sustainable manner [18]. In this context, the principle of the social function of property seeks to guarantee urban justice through the fair distribution of the burdens and benefits of the urbanization process [19].

It was in this scenario of uneven urban development and the emergence of new laws due to the growing demand for more sustainable, fair and democratic cities, that the City Statute [20], the main instrument of Brazilian urban policy, was drafted the constitutional measures on urban policy [21]. Its objective is to order the development of the social functions of the city and urban property, in favor of the collective good, security, citizens' welfare and environmental balance, through 16 guidelines that converge towards a consensus between the agendas urban reform and sustainable development.

Worldwide, the City Statute is highly regarded as being a milestone in the history of urban law, policy and planning for urban land use, the control of development and the enhancement of the social function of property [12,22,23] inspiring instrument for action by national governments [24] and an example of how a large number of stakeholders from different sectors of society have, in adverse circumstances, succeeded in realizing a high quality legal and technical instrument [12,21]. It reflects the view of a company, public managers and legislators from the time of its creation, on the role that the city should play in reducing social inequality, promoting well-being and more democratic management of the city [21,25]. In addition, it was also innovative by explicitly recognizing the right to a sustainable city in Brazil [21].

More recently, the challenges posed by the increasing urbanization experienced by most countries have increased societal demands for more efficient and sustainable urban services, which, in a digital revolution environment, originated and enhanced the concept of the smart and sustainable city [26]. Although discussions about smart cities are not recent, there is no consensus on what a smart city is. Different conceptualizations have been attributed to the term smart city by researchers (Table 1) and governments (Table 2).

**Table 1.** Definitions of smart city by researchers.

| Definitions of Smart City | Authors |
|---|---|
| A city where the Information and Communications Technology strengthens the freedom of speech and the accessibility to public information and services. | [27] |
| A city well performing in a forward-looking way in economy, people, governance, mobility, environment and living, built on the smart combination of endowments and activities of self-decisive, independent and aware citizens. | [28] |
| A city connecting the physical infrastructure, the IT infrastructure, the social infrastructure and the business infrastructure to leverage the collective intelligence of the city. | [29] |
| The use of smart technologies lo make the critical infrastructure components and services of a city—which include city administration, education, healthcare, public safety, real estate, transportation and utilities—more intelligent, interconnected and efficient. | [30] |
| A city to be smart when investments in human and social capital and traditional (transport) and modern (ICT) communication infrastructure fuel sustainable economic growth and a high quality of life, with a wise management of natural resources, through participatory governance. | [31] |
| Smart cities should be regarded as systems of people interacting with and using flows of energy, materials, services and financing to catalyze sustainable economic development, resilience and high quality of life; these flows and interactions become smart through making strategic use of information and communication infrastructure and services in a process of transparent urban planning and management that is responsive to the social and economic needs of society. | [32] |
| Communication technologies into urban management, and use these elements as tools to stimulate the design of an effective government that includes collaborative planning and citizen participation. By promoting integrated and sustainable development, smart cities become more innovative, competitive, attractive and resilient, thus improving lives. | [33] |

**Table 2.** Definitions of smart city by governments (based on [34]).

| Definitions of Smart City | Countries |
|---|---|
| Initially, the concept was only used in a narrow and governmental context especially in relation to environmental, energy and infrastructure issues in terms of how information and communication technologies can improve urban functionality. Subsequently, virtually all other areas of welfare started working with smart city, for example in business development, innovation, citizen involvement, culture, healthcare and social services, where the use of data and digital platforms helps smart new solutions. | Denmark |
| Makes use of opportunities from digitalization, clean energy and technologies, as well as innovative transport technologies, thus providing options for inhabitants to make more environmentally friendly choices and boost sustainable economic growth and enabling cities to improve their service delivery. | Korea |
| City which implements a strategic package of measures to address the most pressing challenges and boost the competitiveness of the area, providing solutions for citizens and entrepreneurs, inter alia such measures which i) do not require substantial maintenance in the long term (save resources); ii) provide more efficient public services (faster, more comfortable, cheaper, e-services, one-stop shop principle); iii) improve overall well-being of society, security and public order; iv) allow timely anticipation and prevention of potential challenges (flood hazards, energy shortages, heat losses, sewer leaks, etc.); iv) do not affect, reduce or eliminate impact on environment; and v) is based on smart development planning, which responds flexibly to the most pressing challenges and development opportunities in the area, identifying existing and potential competitive sectors and promoting their development, as well as providing cooperation between different stakeholders (public administration, entrepreneurs, academics, NGOs, citizens). | Latvia |
| The smart city concept is a holistic approach to cities that uses ICT to improve inhabitants' quality of life and accessibility and ensures consistently improving sustainable economic, social and environmental development. It enables cross-cutting interaction between citizens and cities, and real-time, quality-efficient and cost-effective adaptation to their needs, providing open data and solutions and services geared towards citizens as people. | Spain |
| The concept [of smart city] is not static: there is no absolute definition of a smart city, no end-point, but rather a process, or series of steps, by which cities become more "live able" and resilient and, hence, able to respond quickly to new challenges. | United Kingdom |

Smart cities have been discussed under different approaches, mainly related to understanding the dimensions that characterize them, the drivers that enhance their intelligence and identifying how smart a city is [26], usually through indicators. Aiming to understand the dimensions and characteristics of smart cities, [35] established four main themes: society, economy, environment and governance, which are addressed considering four attributes: sustainability, quality of life, urban aspects and intelligence. [36] proposed a set of factors to understanding smart city initiatives and projects: management and organization, technology, governance, policy, people and communities, economy, built infrastructure and natural environment. [37] identified four key dimensions: economic (GDP, sector strength, international transactions, foreign investment); human (talent, innovation, creativity, education); social (traditions, habits, religions, families); environmental (energy policies, waste and water management, landscape); institutional (civic engagement, administrative authority, elections). [38] established ten main areas: health, effective use of resources, Information and Communications Technology literacy, public administration, regional economics, education, innovative services, culture and recreation, public safety.

Some researchers have sought to establish common sense about the dimensions and characteristics of smart cities from the various interpretations found in the literature [39], summarize these interpretations in three types of approaches: techno centered approach, emphasizing hardware, new technologies and infrastructure; human-centered approach emphasizing social and human capital; and integrated approach that combines both the foregoing qualities. [40] concluded by a set of common multidimensional components, which were categorized into three dimensions: technology, which

encompasses the aspects of hardware and software infrastructures related to the themes physical infrastructure, smart technologies, mobile technologies, virtual technologies and digital networks; people, which encompasses aspects of creativity, diversity and education; and Institutions, which encompasses the aspects of governance, policy, regulations and directives.

In order to identify the main drivers for increasing the intelligence of cities, [26] carried out extensive and detailed bibliographic research, concluding that seven drivers (urban planning, cities infrastructure, sustainability, mobility, public safety, health and public policies) have greater potential for the development of more intelligent and sustainable cities. [41] identified three key elements in the literature: information and communication technologies and smart citizens.

Assuming that every city needs indexes to measure its performance, sets of indicators have also been developed for this purpose. [28] established a set of 74 indicators grouped in 31 factors and six characteristics aiming at the classification of cities in relation to competitiveness, social and human capital, governance, mobility, environment and quality of life. Following the principles set out in ISO 37101, the ISO-37120 established a set of 100 city performance indices grouped in 17 themes (economy, education, energy, environment, finance, fire and emergency, response, governance, health, recreation, safety, shelter, solid waste, telecommunications and innovation, transportation, urban planning, wastewater, water and sanitation) that can be used to monitor the evolution of a city's sustainable development. In the report produced for the European Commission DG Environment by the Science Communication Unit [42], a list of 14 indicator frameworks scalable and easy to implement and 13 tools that may not be as scalable and easy to implement are presented. [43], using a tool for assessing the sustainability of regions, the Ecological Footprint (EF), developed a local approach for using EF to assess the environmental carrying capacity of cities, aiming at more sustainable spatial management. The EF represents the ecological space needed to sustain a given economic system, by accounting for the inflows and outflows of matter and energy from this system, translating them into an equivalent area on land or productive water. All these tools, depending on the results they provide, have an explanatory character, by highlighting good practices to promote them, or simply aim at performance evaluation.

Over time, the engagement of cities with the concept of smart cities has produced three distinct generations depending on how they adopted technology and development [44]: Smart Cities 1.0—technology-driven, in which local managers are persuaded by technology companies to use their solutions in cities that were not prepared to understand how they could affect the quality of life of citizens; Smart Cities 2.0—technology-enabled, city-led, in which the deployment of smart technologies and other innovations happens from the point of view city managers on how it should evolve to improve the quality of life; and Smart Cities 3.0—citizen co-creation, in which citizens have a more active presence in the process of transforming cities.

Reference [45] identified five ways to increase public participation in municipal decision-making and standard-setting: embrace smart cities: encourage the population, the private sector and government organizations to use digital technologies and devices for civic participation and the construction of local public values; cultivate local innovation ecosystems: cities must use the talent and knowledge existing in the local community to implement technologies that meet the needs of these communities; invite public influence: reinvention of the means of public involvement in the decision-making process through new approaches to participatory action and the absorption of new technologies; question data: discuss how and why data is collected and how it is used, in order to prevent violations of people's privacy and civil rights; and design for play and civic imagination: incorporate creativity, experimentation and involvement of the range of actors involved in the conception, design and construction of the smart city, encouraging the inclusion of local values and priorities.

The development of new information technologies has collaborated to improve governmental actions and to engage citizens in the process of building smart cities. In this context, concepts such as e-government, e-participation and e-planning have emerged. E-government can be understood

as the use of technology by the government, mainly web-based applications, to improve access and provision of government information and services to city stakeholders, as well as to bring governments and citizens closer together [46]. According to [47], e-participation is the process of engaging citizens through ICT in policy and decision-making in order to make it participatory, inclusive and deliberative. An e-participation maturity model was presented by [48], based on three-stage approach: e-decision-making, evolving citizens directly in decision processes; e-consultation, organizing public consultations online; and e-information, provision of information via the Internet. Participatory e-planning can be understood as the use of technology to integrate spatial planning approaches, audience participation and visualization techniques [49].

Since the creation of the smart city concept, numerous studies have been developed to understand its characteristics and enhance its evolution. However, when considering the standards that set general guidelines for urban policy, we find that most of them predate the transformations that the concept of smart city provoked the way cities are appropriated and perceived by society. We also find that studies on how these regulations on urban property collaborate to make cities more intelligent and sustainable are scarce.

This work contributes to filling this gap by investigating the main guidelines of the Brazilian City Statute with the potential to contribute to having smarter and more sustainable Brazilian cities, based on the vision of 160 Brazilian professionals who have expertise in the topics addressed by these guidelines.

## 2. City Statute Guidelines

The City Statute is the main instrument of Brazil's urban policy. It is a Brazilian federal law (Law 10,257 of 10 July 2001) that contains sixteen guidelines that guide the actions of the municipalities regarding the development of the social functions of the city and urban property, aiming at environmental balance, the collective good, security and the well-being of citizens. The construction process was participatory, but time-consuming, due to the large number of stakeholders from different sectors of society who participated in its elaboration [50].

Table 3 presents the sixteen guidelines and authors that reference them in works whose themes are related to the guarantee of the right to sustainable cities; democratic management; cooperation between segments of society; city planning; provision of urban and community equipment, transportation and public services; land use ordering and control; integration and complementarity between urban and rural activities; adoption of production and consumption patterns; equity in the distribution of benefits and burdens to the community; adequacy of instruments and public spending; recovery of government investments; protection, preservation and restoration of the natural and built environment; audience between municipal government and population; land regularization and urbanization of areas occupied by low-income population; simplification of urban and environmental legislation; equity of conditions for public and private agents.

These guidelines create a new scenario of new opportunities and obligations for city development management and financing. Among the main advances that these guidelines provide for the evolution of Brazilian cities, we highlight: they regulate the social function when establishing the use of property as of public interest; make the distribution of benefits and burdens arising from the urbanization process more balanced; promote collective well-being and social justice as one of the main obligations to meet the needs of citizens by the government; establish the democratic management of the city through the participation of the population in decisions of public interest; ensure that the population has democratic access to public services and urban facilities; promote the dimensions of sustainability as a fundamental element of spatial planning, guaranteeing the right to sustainable cities; and promote the protection, preservation and recovery of natural and built heritage.

**Table 3.** City Statute guidelines.

| Themes | Guidelines | References |
|---|---|---|
| Guarantee the right to sustainable cities | I – to guarantee the right to sustainable cities, understood as the right to urban land, housing, environmental sanitation, urban infrastructure, transportation and public services, employment and leisure, for current and future generations. | [51–53] |
| Democratic management | II – democratic administration by means of participation by the population and the representative associations of the various sectors of the community in the formulation, execution and monitoring of urban development projects, plans and programmers. | [54–56] |
| Cooperation between segments of society | III – cooperation between governments, the private sector and other sectors of society in the urbanization process, to satisfy the social interest. | [57–59] |
| City planning | IV – planning of the development of cities, of spatial distribution of the population and of the economic activities of the municipality and of the territory under its area of influence, in order to avoid and correct distortions caused by urban growth and its negative effects on the environment. | [51,57,60] |
| Provision of urban and community equipment, transportation and public services | V – provision of urban and community equipment, transportation and public services that are appropriate to the interests and needs of the population as well as reflecting local circumstances. | [61–63] |
| Land use ordering and control | VI – ordering and control of land use, in order to avoid: a) the improper use of urban real estate; b) the proximity of incompatible or inconvenient uses; c) the parceling of land, construction or excessive or improper use with regard to urban infrastructure; d) the installation of developments or activities that could become hubs that generate traffic, with no prevision for corresponding infrastructure; e) the speculative retention of urban real estate, resulting in its underutilization or no utilization; f) the deterioration of urbanized areas; g) pollution and environmental degradation. | [22,64,65] |
| Integration and complementarity between urban and rural activities | VII – integration and complementarity between urban and rural activities, taking account of the social-economic development of the municipality and the territory under its area of influence. | [66–68] |
| Adoption of production and consumption patterns | VIII – the adoption of production and consumption patterns related to goods and services and of standards of urban expansion compatible with the limits of environmental, social and economic sustainability of the municipality and of the territory under its area of influence. | [69–71] |
| Equity in the distribution of benefits and burdens to the community | IX – the fair distribution of the costs and benefits resulting from the urbanization process. | [19,72,73] |
| Adequacy of instruments and public spending | X – the adaptation of economic, taxation and financial policy instruments and public expenditure to suit the goals of urban development, in order to give priority to investments that generate general well-being and enjoyment of the assets by different social segments. | [74–76] |
| Recovery of government investments | XI – recovery of government investments that have led to appreciation in the value of the urban property. | [19,77,78] |
| Protection, preservation, and restoration of the natural and built environment | XII – protection, preservation and recovery of the natural and built environment, and of the cultural, historic, artistic, landscape and archaeological heritage. | [52,79,80] |
| Audience between municipal government and population | XIII – public hearings involving municipal governments and members of the population interested in the processes of execution of developments or activities with potentially negative effects on the natural or built environment, the comfort or safety of the population. | [56,81,82] |
| Land regularization and urbanization of areas occupied by the low-income population | XIV – tenure regularization and urbanization of areas occupied by low-income populations through the establishment of special urbanization, land use, land occupation and building norms, taking due account of the socio-economic situation of the population and environmental norms. | [83–85] |
| Simplification of urban and environmental legislation | XV – simplification of the legislation concerning subdivision, land use, occupation and building regulations, in order to permit cost reductions and increase the supply of lots and housing units. | [78,86,87] |
| Equity of conditions for public and private agents | XVI – equality of conditions for public and private agents in the promotion of developments and activities related to the urbanization process, serving the social interest. | [88–90] |

## 3. Materials and Methods

The main research question of this study was "what are the main guidelines of the City Statute with the potential to contribute to having smarter and more sustainable cities in Brazil?" To answer this question, we designed an approach in three steps: bibliographic research, survey of expert's opinions and data analysis (Figure 1).

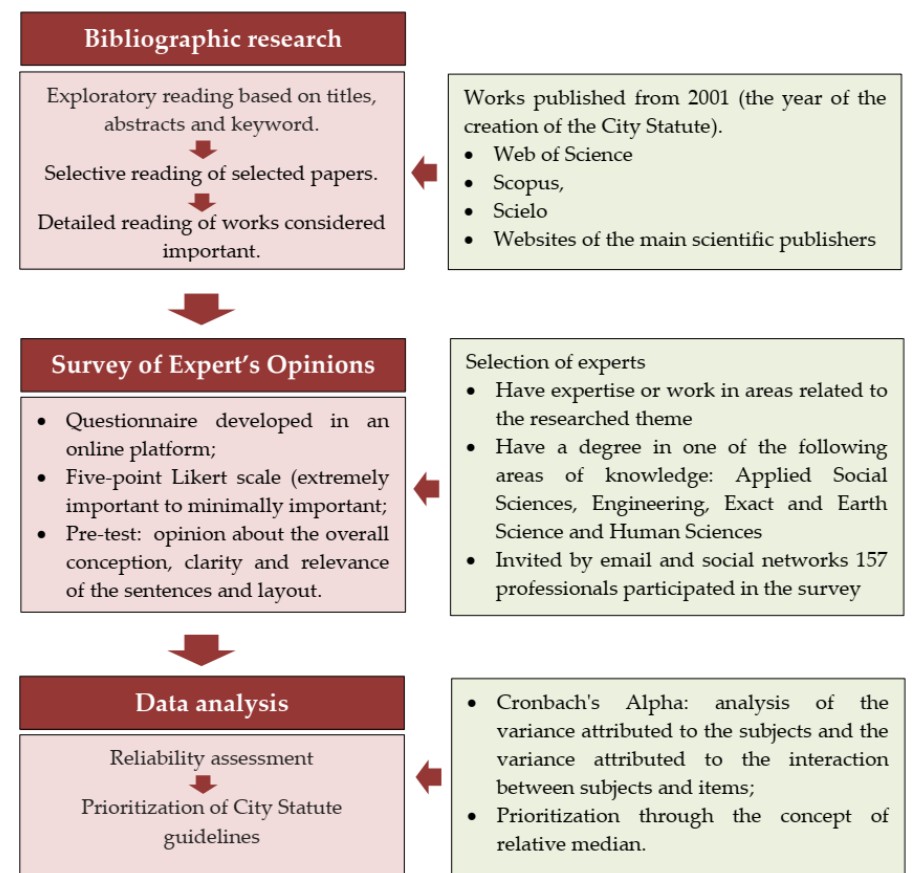

**Figure 1.** Synthesis of the adopted methodology.

### 3.1. Bibliographic Research

Comprehensive and detailed bibliographic research was carried out on the Web of Science, Scopus, Scielo and on the websites of the main scientific publishers, covering works published from 2001, the year of the creation of the City Statute.

For the search for articles in the English language, we decided to consider all articles that contain the keyword City Statute. Considering a large number of works on the subject in Portuguese, the search for articles in Portuguese was more selective, combining the keyword City Statute with keywords that were chosen in order to consider the topics covered in the Statute guidelines. We use the "and" connector so that the works identified always addressed the topics relating them to the Statute. The keywords used were: guarantee of the right to sustainable cities; cooperation between the segments of society; cooperation in the urbanization process; provision of urban and community equipment; public services appropriate to the population; demographic management; sustainable cities; urbanization process; town development planning; master plan; land use ordering and control; urban and rural activities; environmental sustainability; production and consumption of goods and services; benefit and burden distribution; economic, tax, financial and public expenditure policy instruments; public spending; recovery of public investment; real estate valuation; natural and built environment; environmental preservation, protection and recovery; ambiental degradation; municipal

hearing, population, enterprise, security; municipal public hearing; land reclamation and urbanization of occupied areas; special urbanization norms; simplification of land parceling legislation.

The bibliographic search was carried out taking into consideration mainly the recommendations of Webster and Watson (2002) [26,91], and the Preferred Reporting Items for Systematic Reviews and Meta Analyzes—PRISMA [92]. Initially, we performed exploratory reading based on titles, abstracts and keywords, to select the works that correlated with the researched theme. Then we performed selective reading of the selected works, and those that did not contain information relevant to the research questions were excluded.

As a result, we obtained 2617 articles of which 590 were excluded because they appeared duplicate. In the remaining 2027 articles, we performed an exploratory reading of titles and abstracts, and 1782 papers were discarded due to the following exclusion criteria: abstracts that were not relevant to the researched theme, articles published in journals without peer review system and articles that were not available in their journal completeness.

Then, we performed selective reading in the remaining 245 works aiming at proving our perception when reading the abstracts, and 86 articles were excluded due to the following exclusion criteria: articles that were not original, results that did not contribute to the researched theme, methodology that was not sufficiently presented and results that were not sufficiently supported by the methodology.

Finally, we made a detailed reading of the 159 remaining works and prepared a spreadsheet containing the most relevant excerpts for the researched theme, having effectively taken advantage of a total of 136 works. Figure 2 summarizes the literature search through the PRISMA flowchart.

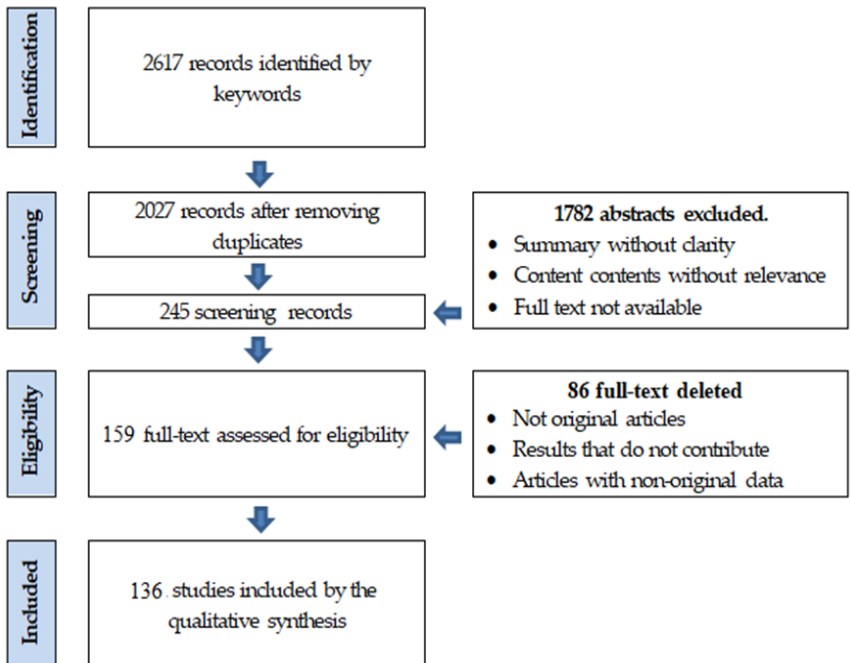

**Figure 2.** Literature search from the Preferred Reporting Items for Systematic Reviews and Meta Analyzes (PRISMA) flowchart.

### 3.2. Survey of Experts' Opinions

The survey was conducted through a questionnaire developed on an online platform, containing two sections: demographic data and questions addressing the importance of city statute guidelines for increasing city intelligence. Respondents expressed their opinion according to a five-point Likert scale, ranging from extremely important to minimally important, with guidelines presented at random so that the order in which they appeared did not influence responses.

The professionals were selected considering the following inclusion criteria: have expertise or work in areas related to the researched theme and have a degree in one of the following areas of knowledge:

Applied Social Sciences, Engineering, Exact and Earth Science and Human Sciences. In Brazil, the structuring of these knowledge areas is carried out by the Coordination for the Improvement of Higher Education Personnel, based on the affinity of objects, cognitive methods and instrumental resources of the training areas, as follows [26]:

Applied social sciences: It is formed by areas of interdisciplinary training that deal with aspects related to public and private administration, architecture, urbanism and design, accounting and tourism, communication and information, law, economics, urban and regional planning/demography and social work.

Human Sciences: Focuses mainly on the relationships of human beings with history, their beliefs and the local/temporal space that connect them, encompassing the following areas of formation: anthropology, archeology, political science and international relations, the sciences of religion and theology, education, philosophy, geography, history, psychology and sociology.

Engineering: Focuses on the various branches of technology that through methods, techniques and scientific vision, make it possible to solve problems and materialize ideas in reality, satisfying human needs. They encompass all engineering degrees.

Exact and Earth Sciences: It is formed by areas of formation based on physical-mathematical calculations, such as astronomy, computer science, statistics, physics, geosciences, mathematics and chemistry.

For the pre-test, we used a printed questionnaire to give professionals an opinion about the overall conception, clarity and relevance of the sentences and layout.

We used email and social networks to invite 536 professionals to participate in the survey, of which 157 participated in expressing their opinions from 05/24/2019 to 06/08/2019. To access the experts, we used four strategies:

- To invite participants from the Industry 4.0—Smart city group of 226 experts and scholars on the subject;
- To invite the participants of the group "UFF Engenharia", composed of 35 engineers graduated from UFF in 1984, with vast experience in projects and urban intervention;
- To invite participants to the "poscivil@googlegroups.com" group of 220 engineers, architects and administrators working in the area;
- To use the authors' personal relationships to invite fifty-five professionals (engineers, architects, lawyers, teachers, administrators, public and private managers), all with proven expertise in the researched subject.

*3.3. Data Analysis*

The reliability of the questionnaire and respondents was assessed by measuring the variance of the responses of each item and the variance of the responses of each respondent (Martins et al., 2011) [93] was performed using Cronbach's alpha. [93] because it takes into account the variance attributed to the subjects and the variance attributed to the interaction between subjects and items, in order to evaluate the average of correlations between items that are part of the questionnaire and the measure by which the measured factor is present in each item [26,94].

The guidelines were prioritized through the concept of relative median [26], which allowed driver hierarchy in each semantic of the Likert scale. Taking as an example in Figure 3, the median (equal to four) of the first line is much closer to the frequency represented by the number three. In the second row, the median shifts to the right as you add cells at the frequency represented by the number five. Although in both lines we have a median of four, the guideline represented by the second line can be interpreted as more important, as it received five more ratings and maintained the other frequencies.

| 1 | 2 | 2 | 3 | 3 | 3 | 3 | 3 | 3 | **4** | 4 | 4 | 4 | 4 | 4 | 5 | 5 | 5 | 5 | | | | | | | | | | |
|---|---|---|---|---|---|---|---|---|---|---|---|---|---|---|---|---|---|---|---|---|---|---|---|---|---|---|---|
| 1 | 2 | 2 | 3 | 3 | 3 | 3 | 3 | 3 | 4 | 4 | 4 | 4 | 4 | **4** | 5 | 5 | 5 | 5 | 5 | 5 | 5 | 5 | 5 | 5 | 5 | 5 | 5 |

**Figure 3.** Example of the median position.

For the calculation of the relative median we generalize the original formula, allowing the calculation of the relative median for any Likert scale, any of the intervals of this scale and for fractional medians:

$$
\text{Rm} = \begin{cases}
1 + \frac{Pr}{j_1} & for \ m = 1 \\
m + \frac{Pr - \left( \sum_{i=1}^{m-1} j_i + 1 \right)}{j_m} & for \ 2 \leq m < N \ and \ m = integer \\
m + 0.5 & for \ 1 \leq m < N \ and \ m = Fractional \ number \\
N & for \ m = N
\end{cases}
$$

where: Rm is the relative median, $m$ is the median, $Pr$ is the relative position of the median, $j_i$ is the number of respondents who were assigned a semantic classification of "$i$", and $N$ is the maximum value of the Likert scale used.

## 4. Results and Discussion

Initially, we calculated the Cronbach's alpha, whose value equal to 0.89 confirmed the reliability of the questionnaire and the data. Then, we identified the profile of respondents from the demographic data of the first section of the questionnaire considering their area of training and length of professional experience (Figure 4). For the four training areas, at least 66% of respondents had more than ten years of experience.

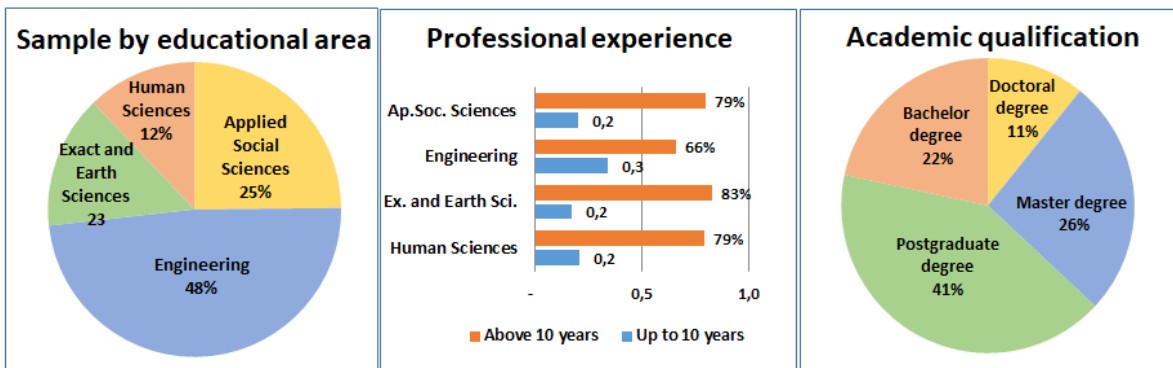

**Figure 4.** Demographic data.

Based on the judgment of the specialists from each training area, the guidelines were classified by the relative median (Figure 5). Figure 6 presents the same classification for all respondents.

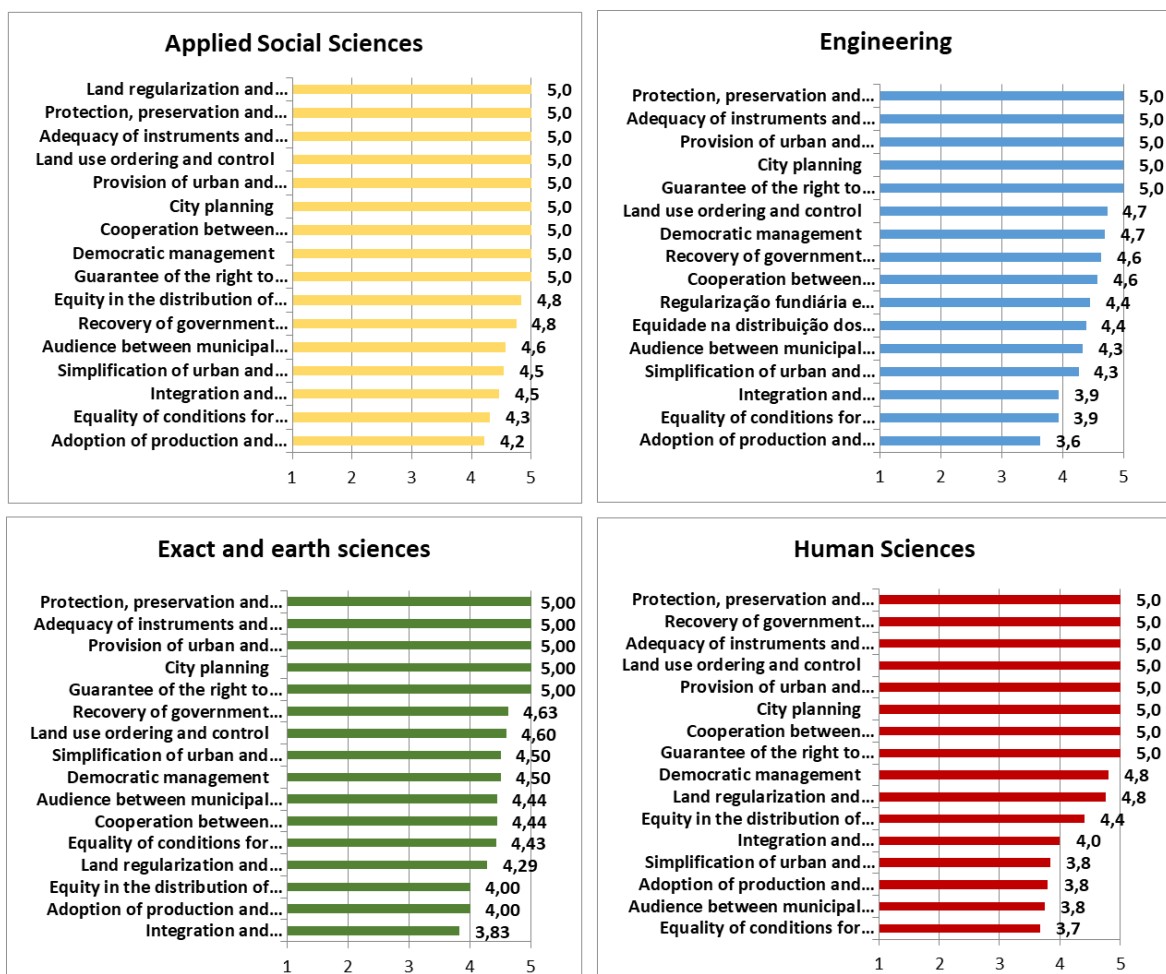

**Figure 5.** Guidelines ranked by the relative median for the four areas of knowledge.

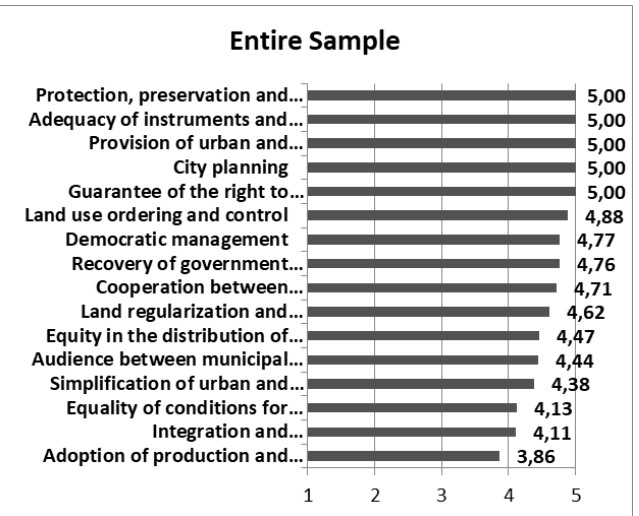

**Figure 6.** Guidelines ranked by the relative median based on the total respondents.

Figures 5 and 6 show that all guidelines were considered important by the experts since relative medians were greater than 3.0.

Table 4 lists the guidelines that were rated by experts as "extremely important" from the relative median.

**Table 4.** Guidelines ranked as "extremely important".

| Guidelines | Applied Social Sciences | Engineering | Exact and Earth Sciences | Human Sciences | Entire Sample |
|---|---|---|---|---|---|
| Guarantee the right to sustainable cities | ➎ | ➎ | ➎ | ➎ | ➎ |
| City planning | ➎ | ➎ | ➎ | ➎ | ➎ |
| Provision of urban and community equipment, transportation and public services | ➎ | ➎ | ➎ | ➎ | ➎ |
| Adequacy of instruments and public spending | ➎ | ➎ | ➎ | ➎ | ➎ |
| Protection, preservation and restoration of the natural and built environment | ➎ | ➎ | ➎ | ➎ | ➎ |
| Land use ordering and control | ➎ | | | ➎ | |
| Cooperation between segments of society | ➎ | | | ➎ | |
| Land regularization and urbanization of areas occupied by low-income population | ➎ | | | | |
| Democratic management | ➎ | | | | |
| Recovery of government investments | | | | ➎ | |

Ten guidelines were considered "extremely important" by the experts (Table 4), and only five were considered "extremely important" for all areas of training and for the entire sample, which we will call Top 5 Guidelines: guarantee of the right to sustainable cities; city planning; provision of urban and community equipment, transportation and public services; adequacy of instruments and public spending; protection, preservation and restoration of the natural and built environment.

This set of five guidelines was considered to be the most important and should be prioritized by stakeholders for decision making.

Figures 6 and 7 show the behavior of the guidelines when evaluations by training areas are compared with evaluations performed for the entire sample.

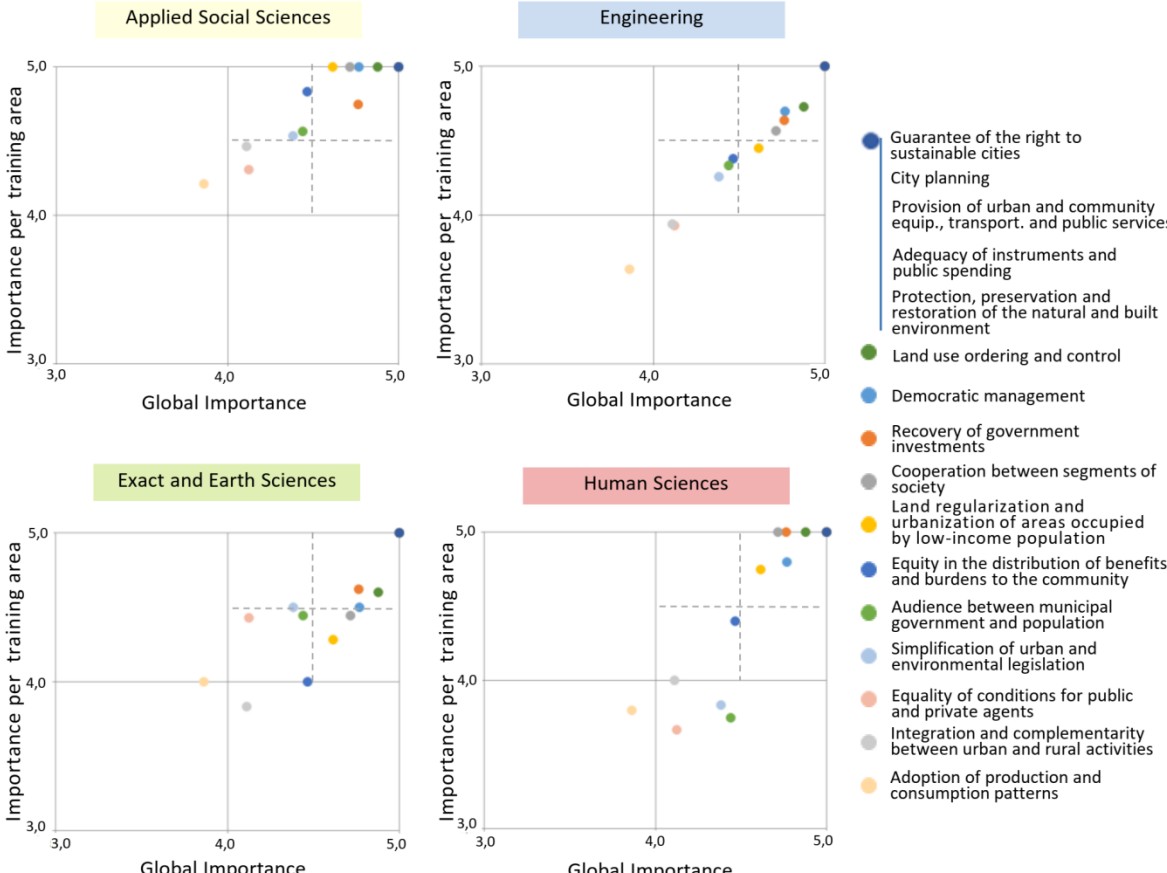

**Figure 7.** Guidelines' behavior by training areas, related to the whole sample.

In Figures 7 and 8, the guidelines considered important were those that had their relative medians ranging from 4.01 to 4.99, which in descending order were: land use ordering and control; democratic management; recovery of public investment; cooperation between the segments of society; land regularization and urbanization of areas occupied by low-income population; equity in the distribution of benefits and burdens to the community; audiences between the municipal government and the population; simplification of urban and environmental legislation; fairness of conditions for public and private agents; integration and complementarity between urban and rural activities. The guideline "ordering and control of land use" obtained a relative median of 4.88 and can be considered borderline, standing out as a guideline "extremely important" for the applied social sciences knowledge groups and for the human sciences group. Conversely, to the guideline "isonomy of conditions for public and private agents" and the guideline "integration and complementarity between urban and rural activities" obtained a relative median of 4.13 and 4.11 respectively, were considered of low priority by two of the four knowledge groups consulted. Only the guideline "adoption of production and consumption patterns" had its relative median between 3 and 4, that is, it was evaluated as "important" but not a priority.

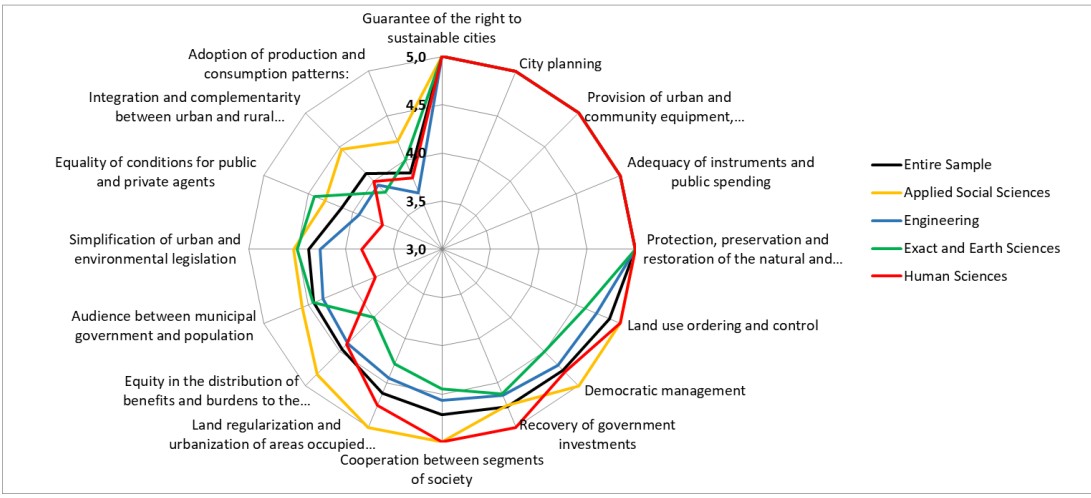

**Figure 8.** Drivers' behavior.

With regard to guideline city planning, it must be considered in the context that Brazil, in addition to being the largest urbanized nation in Latin America [95], has also experienced rapid growth in its cities, which has led to problems such as reduced quality of life caused by traffic congestion, increased pollution and increased social inequality [96].

Discussions about how new approaches to urban planning can enable prosperity and intelligence in metropolitan areas, which has occurred in several countries [96,97] in Brazil, have focused mainly on capitals and large urban centers, which are the main ones more likely to become smarter. This phenomenon has been identified by some researchers [96,98] in relation to large cities, since, on the one hand, these cities, due to their high demographic density, facilitate the flow of knowledge and social interactions, which enhances the emergence of new ideas and innovation, and have also solved local public transport and infrastructure systems more effectively, making them better suited to the implementation of smart city concepts; on the other hand, disorderly growth produces problems related to mobility, security, energy consumption and infrastructure deficiencies, which while making them less intelligent, also potentially make them more interested in implementing smart city concepts as a way to address these problems.

When considering city planning in a smart city context, it involves the management of territories by defining the priorities that operationalize public policies, aiming at urban environmental quality and well-being in a manner connected with all areas of the city. [26] shows the convergence between

these concepts and those of the guideline. Several authors [91,99–108] cite these characteristics as important for increasing the intelligence of cities.

With regard to the "guarantee of the right to sustainable cities" and "provision of urban and community equipment, transportation and public services" guidelines, it is important to keep in mind that the right to the city is diffusive, collective and indivisible in nature and should be considered from the point of view of guaranteeing and promoting human rights, which implies fair, inclusive, democratic and sustainable cities in which every citizen is given the right to live, use and participate in their production [109].

In the Statute, this means the right to urban land, housing, environmental sanitation, urban infrastructure, transportation and public services, work and leisure, appropriate to the interests and needs of the population and local characteristics, for those present and future generations. Several authors [100,108,110–115] cite these characteristics as important for increasing the intelligence of cities.

Regarding the adequacy of instruments and public spending guideline, considering that public spending is the means by which public administration finances its policies and provides public goods and services, a significant portion of public managers in Brazil do not seem to worry about whether the funds raised, mainly through taxes, actually achieve the goal of providing the well-being of society by meeting their demands and needs [116], since indicators are not usually used to measure how public expenditure has achieved results [117], even considering that public expenditures serve several purposes and that part of the results of public service actions are intangible, which makes cost/benefit analysis difficult.

However, by providing for the possibility for the population to participate in the process of setting investment priorities for the municipality, the City Statute contributes to improving the efficiency and transparency of the use of public resources.

This guideline enhances investments that generate general welfare and the enjoyment of assets by different social segments, through economic, tax and financial policy instruments and public spending. Several authors [108,111,114,115] cite such characteristics as being important for increasing the intelligence of cities.

With regard to the guideline on the protection, preservation and restoration of the natural and built environment, including cultural, historical, artistic, scenic and archaeological heritage, to consider public heritage is that which consists of the movable and immovable property of persons governed by public law [118]. But the public adjective can refer to what belongs to the state as well as to everything that belongs to the community, but which is in the custody and supervision of the state. On the other hand, the idea of equity is not limited to economically valued assets, but also extends to other assets, even if they are devoid of economic value.

In this context, with regard to the environment, it is important to note that the actions and strategies that are part of the urban policy are inextricably linked to environmental protection. If the objective of urban policy, as a result of the urbanization process, is to organize urban spaces, it could not ignore that the ecologically balanced environment is one of the most important ways of enabling a better quality of life for the city's inhabitants and users.

With regard to cultural, historical, artistic, landscape and archaeological heritage, the main objective is to prevent real estate speculation and other private interests from damaging such values, especially as destruction often seems irreversible. Heritage preserved in the past can serve as inspiration for future urban development, as the present, past and future are connected by the smart city [101].

Conservation and preservation of these heritage sites have been considered an important theme in several areas related to smart cities, such as public planning, urban development, sustainable development [119–121]. Several authors [106,122–125], in different approaches and contexts, have considered the themes present in this guideline to be important for increasing the intelligence of cities. An example is the work of [126], which establishes the nexus between smart technologies, heritage conservation and progress towards inclusive and sustainable cities and communities.

In the work of [26], seven drivers were identified as being the most important for increasing city intelligence, of which four of these drives coincided with the five most important guidelines identified in the survey. From this result, we relate the City Statute guidelines classified as "extremely important" to the smart and sustainable city drivers (Table 5).

**Table 5.** Comparison between major smart cities drivers and the guidelines.

| Main Drivers [26] | Main Guidelines |
|---|---|
| **Sustainability**: Efficient management of natural resources contributes to raising citizens' quality of life for current and future generations. Social, economic and environmental sustainability are strategic vectors for smart cities. | **Guaranteeing the right to sustainable cities**: Guaranteeing the right to urban land, housing, environmental sanitation, urban infrastructure, transportation, public services, work and leisure for present and future generations. |
| | **Protection, preservation and restoration of the natural and built environment, cultural, historical, artistic, landscape and archaeological heritage**: Guaranteed for the protection, preservation and restoration of the natural and built environment and material and non-material works that reflect creativity and values. Society and urban history, including through the spatial distribution of the population and economic activities, thus avoiding the distortions and negative effects of urban growth and real estate speculation. |
| **Urban planning**: Territorial management through tools and indexes, including urban environmental quality, air quality and well-being. This connects with all areas of the city because, for developing cities, planning is a key tool in defining the priorities that operationalize public policies, enabling cities to become smarter and more sustainable. | **City planning**: The planning must contain guidelines and norms that regulate the development of cities, the spatial distribution of the population and the economic activities of the municipality and the territory under its area of influence, being the basic instrument of the urban development and expansion policy. It should also consider the development of strategies based on environmental sustainability, thus fulfilling its social and environmental function. |
| **City Infrastructure**: Management of basic storm water networks, sanitation and water and sewage services. These should be managed as living systems, with efficient operation and management, requiring large-scale management to provide at least minimal finite resource sustainability. | **Supply of urban and community equipment, transport and public services**: Guarantee of the existence of urban and community equipment, consisting of a set of goods, physical spaces and buildings of public utility that provide the material support and provision of basic health services, education, recreation, sports and other needs of society related to health, welfare and exercise of citizenship. |
| **Public policies**: The planning and development of public policies in favor of an intelligent city, because municipal administrations are the entities that rely heavily on local policies to manage projects, actions and services that, by involving several actors, can sometimes seem conflicting. | **Adequacy of instruments and public expenditures**: Adequacy of economic, tax and financial policy instruments and public expenditures to the objectives of urban development, so as to privilege investments that generate general welfare and the enjoyment of assets by different social segments. It also includes the city's democratic management with popular participation through communities, movements and societal entities, aiming at the definition of public policies and the approval of legislation authorizing public spending. |

By comparing smart city drivers with the main guidelines, we can state that the City Statute is an instrument that seeks: the sustainability of cities by "guaranteeing the right of sustainable cities" and "protecting, preserving and restoring the natural environment and built"; urban planning in determining "city planning"; the improvement of the city's infrastructure by providing for "provision of urban and community facilities, transportation, and public services" and public policies for "adequacy of public instruments and expenditures".

The results show that the group of the five most important guidelines is related to city governance.

We consider that governance is a management tool that enhances the sustainable development of cities by articulating stakeholder interests, transparency and equity in order to resolve conflicts throughout the territory and to implement smart solutions within the participatory process of citizens. In this sense, it is a theme related to the capacity of articulation and cooperation between different actors of a company, for the discussion of issues of common interest. Some researchers [127–129] have this same view.

The way the City Statute contributes by defining guidelines that regulate urban policy with participatory planning and outlines the social function of property is closely intertwined with governance processes.

However, we consider that the results obtained may have been influenced by the Brazilian reality, political and financial crises that periodically intensify and deteriorate public services and infrastructure of cities, leading to the perception of the absence of planning and management.

From the promulgation of the law that instituted the City Statute, there was a movement of municipal governments aiming at its operationalization [130]. In this sense, there was, for example, the adoption of inclusive democratic management initiatives in territorial planning [130], the expansion of society's participation in important decisions on municipal policies and actions, such as public consultations for approval of plans, projects and standards [131]; the use of instruments aimed at the fair distribution of the burdens and benefits of urbanization [19]; the dissemination of master plans containing the instruments provided for in the Statute [132].

Although the five main guidelines contribute to having smarter and more sustainable cities in Brazil, when considering the 5568 Brazilian municipalities, it appears that there is still much to be done, as several factors hinder the evolution of a significant portion of Brazilian cities, among the which we highlight:

(a) The great disparity of resources and infrastructure between cities means that a significant portion does not have the capacity to implement the concepts of smart cities, and those that started this process are in the situation highlighted by [45], to coexist, each in its own evolution phase (Smart City 1.0, 2.0 and 3.0).

(b) Even today conditions persist that some researchers [95,133] have found, that the urban expansion planning of the emerging metropolises of Latin America and, consequently of Brazil, was strongly influenced by the models and philosophies of existing urbanism in Europe and North America, which influenced generations of architects and city planners, whose ways of thinking and producing the design of Brazilian cities, allowed barriers, especially invisible ones, to continue separating the richest areas from the most poor, often without basic infrastructure and services.

(c) Several researchers [23,25,130,134–136] have found that in Brazil, as in most of the cities in developing countries, informal housing with uneven spaces, insecure land tenure, poor infrastructure and mobility and situations of social and political vulnerability still persist.

(d) In many municipalities the situation identified by [130] is present, that the active participation of citizens in decision-making processes is hampered by the lack of knowledge on how to claim, dialogue, appeal and expose local needs, and by collusion city halls with dominant class interests;

(e) Some municipalities have considered the guidelines superficially, without interpreting them considering the reality of the municipality [130];

(f) Some cities are managed without planning that has development goals compatible with the instruments of economic, tax and financial policy;

(g) The disparity between the municipalities regarding access to technology hinders transparency and dissemination of information, as well as communication, access and participation of citizens.

These structural problems make it difficult to increase the intelligence of Brazilian cities. This is not to say that the City Statute has not made significant progress in the way public management operationalizes the city's functions since prior to the City Statute policies aimed at access to basic services and inequality reductions were sporadic.

In this context, the results of the research are fully justifiable, since the five guidelines evaluated as most important are responsible for: stimulating the development of cities, spatial distribution of the population and the economic activities of the municipality and the territory, in order to avoid and correct distortions caused by urban growth and its negative effects on the environment; to guarantee the right to sustainable cities, understood as the right to urban land, housing, environmental sanitation, urban infrastructure, transportation and public services, employment and leisure, for current and future generations; to protect, preserve and recover the natural and built environment, and cultural, historic, artistic, landscape and archeological heritage; to provide urban and community equipment, transportation and public services that are appropriate to the interests and needs of the population as well as reflecting local circumstances; to adapt economic, taxation and financial policy instruments and public expenditure to suit the goals of urban development, in order to give priority to investments which generate general well-being and enjoyment of the assets by different social segments.

However, some important aspects related to the participation of the main stakeholders in the operationalization process of the five main Statute Guidelines need to be considered from the reality of Brazilian cities, in order for this process to be more effective:

(a)  Managers and public servants involved in the administrative and operational management of the city: We envision four fundamental actions: the first and perhaps most important is to understand and address the concepts, benefits and difficulties of implementing smart cities considering the local reality; the second is to incorporate this knowledge into urban planning and territorial management and public services; the third is to bring citizens closer to the decision-making process, not only to comply with the requirements of Brazilian laws; and, finally, to increase the use of technology as a facilitator of these actions;

(b)  Technology supply companies: The functioning of the city requires a series of public services that can benefit from technology to improve its performance and capacity to meet the demands and needs of citizens. A significant portion of the existing technology can be used without the need for adaptations, this situation is the one that normally presents the best cost-benefit ratio for companies. However, the commitment of these companies to the development of solutions that are more appropriate to local realities is also fundamental. Another important issue refers to the expansion of the role played by these companies, from simply providing technology to a partner and engaged in the process of transforming the city;

(c)  Educational and research institutions: when playing the role of building and disseminating knowledge and practices, they are fundamental for the identification and systematization of the local reality and the development of new approaches that enable the expansion of existing knowledge and practices about smart cities and the adaptation of this knowledge and practices to the local reality. Another important action concerns the formation of local culture on the concepts and benefits of smart cities based on the dissemination of knowledge and the training of professionals. It is also essential to participate in the citizen engagement process, through actions that make it possible to increase their awareness of their rights and responsibilities;

(d)  Citizens and organizations representing citizens: Community participation is essential for compliance with the guidelines to be more effective and meet their needs. In this sense, it must be better able to demand compliance with the rights established in the guidelines, more participatory when demanded by the public power, such as, for example, during the public consultations required by Brazilian law, and more proactive, when demanding and presenting contributions to public power.

## 5. Conclusions

Regulation of urban property use has been a fundamental instrument for cities to develop, considering the collective good, the welfare of citizens and sustainable development.

Although issues intrinsically related to the regulation of urban property use have been the subject of studies related to smart and sustainable cities, the vast majority of the norms that establish general guidelines for urban policy predate the transformations that the smart city concept has caused in the way that cities are appropriated and perceived by society, which justifies the scarcity of studies on how these urban property regulations collaborate to make cities smarter and more sustainable.

This work contributes to filling this gap by investigating the main guidelines of the City Statute, the main instrument of Brazilian urban policy, which have the greatest potential to contribute to having smarter and more sustainable cities.

The results show that the 16 guidelines were evaluated as important for increasing city intelligence, of which five were considered to have the most priority, and these five were related to city governance.

However, we emphasize that these guidelines must be considered within the reality of each municipality because the priorities of society are influenced by the context in which they are inserted. Brazilian cities have their own characteristics, such as government profile, financing

capacity, socio-environmental culture, citizen participation, etc. Thus, the perception of problems and the search for solutions will be different from municipality to municipality.

Given that cities in various countries are experiencing similar situations, especially those in Latin America, these five guidelines also have the potential to contribute to the increase in intelligence of these cities, provided their specificities are considered.

Considering the scenario of resource scarcity in Brazilian cities, these five guidelines help governors to direct their efforts to the highest priority, since for cities to evolve to become smarter, structural problems are fundamental to be solved.

The present study has some limitations. The first is inherent in all bibliographical research, from which some important contribution has escaped our analysis. The second is that for the prioritization of the guidelines, the research was based only on the evaluations of Brazilian experts, which may have been influenced by the Brazilian reality. In this context, generalizations must consider local realities. However, it is important to emphasize that the realities experienced in Brazilian cities are present in most underdeveloped and developing countries.

The City Statute points and regulates the way, but it is innovative and regionalized solutions that will make the statute fully contribute to smarter cities.

**Author Contributions:** Conceptualized the study, C.A.P.S., E.G.L. and A.L.A.G.; C.A.P.S., E.G.L., Contributed to the methodology, C.K.C. and A.N.H.; Contributed the software C.A.P.S., E.G.L., E.G.V. and A.W.A.H.; Formal analysis was performed by C.A.P.S., E.G.L., C.K.C., A.N.H. and A.W.A.H.; Data curation was performed by C.A.P.S., E.G.L., A.W.A.H. and A.N.H.; Validation was done by C.A.P.S., E.G.L., C.K.C., A.L.A.G. and A.N.H.; The original draft was written by C.A.P.S. and E.G.L.; Review and editing were done by C.A.P.S., E.G.L., C.K.C., E.G.V., A.L.A.G., A.W.A.H. and A.N.H. All authors have read and agreed to the published version of the manuscript.

**Funding:** This research received no external funding.

**Acknowledgments:** The authors would like to thank Fluminense Federal University, Brasil, for funding the research reported in this paper. The authors would like to thank all the experts who answered the survey. The authors also express their gratitude to the editor and anonymous reviewers for comments and suggestions.

**Conflicts of Interest:** The authors declare no conflict of interest.

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
