# Peer review of "Smart and Sustainable Cities: The Main Guidelines of City Statute for Increasing the Intelligence of Brazilian Cities"

_sustainability, doi:10.3390/su12031025_

Round 1

Reviewer 1 Report

The title of manuscript isn’t relevant. It suggests that the reader will receive universal information for use on a global scale, but the article contains information useful only locally, in the realities of Brazil.

Introduction should better organize the study from broad world (global) context to Brazil case studies. Authors didn’t review carefully the current global/international state of the research in the field of sustainable smart city and they didn’t refer the key international publications in this theme. As a consequence, important problems and information are missing, such as:

Development of sustainable smart city concept and its definitions, Generation of smart cities (from Smart Cities 1.0: Technology Driven, to Smart Cities 3.0: Citizen co-creation) Examples of normative references from different countries (e.g. ISO 37101, Sustainable development in communities — Management system for sustainable development — Requirements with guidance for use or ISO 37120:2018(en), Sustainable cities and communities — Indicators for city services and quality of life very popular organization the problems of smart cities into six building blocks, such us: „smart people”, „smart economy”, „smart mobility”, ‘smart environment”, „smart living”, „smart governance” (Giffinger, R. and Haindlmaier, G. (2010).

The research concept is not appropriate. It is based on an analysis of 16 guidelines copied from the document published in 2010, i.e. 10 years ago (Rodrigues, E.; Barbosa, B.R. The City Statute of Brazil: A Commentary (2010). Consequently, the proposed set of guidelines weakly addresses important latest global problems in the aspect of development of human sustainable smart cities, such as:

the rise in the number of urban dwellers and aging population, urban sprawl consumption of energy resources, environmental pollution (in particular the problem of smog and waste management), threat to cities posed by extreme weather phenomena associated with global warming (e.g. such as: heat waves, hurricane winds, floods), rising share of digital technologies supporting the functioning of cities and having a greater impact on the lives of their inhabitants.

Reviewer 2 Report

The paper presents very interesting research which fits to the scope of the journal, however, it has some failures which should be improved. I present comments which should be considered and corrected before publishing.

Please follow the structure of the abstract presented in the template of the journal: (1) Background: Place the question addressed in a broad context and highlight the purpose of the study; (2) Methods: Describe briefly the main methods or treatments applied; (3) Results: Summarize the article's main findings; and (4) Conclusions: Indicate the main conclusions or interpretations. In submitted paper methodological elements are missing in the abstract, but also other parts could be better balanced. Paragraphs should introduce the readers to wider concepts and should build consistent part of the story. Please avoid paragraph which contain only one sentence (lines: 29-30, 42-46, 117-119, 120-122, 123-125, etc.) and consider combining some other paragraphs. Please give a source to the statement in lines 42-46. New paragraph opens new idea. Therefore, the opening sentence in the paragraph can not be as follows: “It was from the 1980s that in Latin America these movements (…)”. Which movements? The situation would be different if proper sentences would constitute one paragraph, so such corrections should be done after considering my 2 comment. Line 71: why the word “Government” is written with a capital letter? Lines 68-74: I find it difficult to follow such complex sentence, especially that previously sentences were focusing on small part of information. It does not seem consistent. The literature review in the introduction is quite biased and concentrate only on publications from Latin America. If the intension is to present the wider context of the research it might be useful to consider referring also to papers from other parts of the world too. For instance while discussing tools to control urban sustainability it might be interesting to refer to concept of one of the most popular index for cities – ecological footprint developed by Global Footprint Network (e.g. The application of ecological footprint and biocapacity for environmental carrying capacity assessment: A new approach for European cities [in:] Environmental Science and Policy). Line 104: (Guedes et al., 2018) appear probably by coincidence. Please delete it. Similarily later in line 165-166. References according to MDPI style are reflected by numbers in square brackets. Section 1 should be concluded with the aim of the research like it appeared in the abstract. Figure 1 should be removed from section 3.2 to 3.1 to avoid misunderstanding. Moreover, please improve the quality of the figure and make sure to avoid green underline of “full-text”. Line 245: I suggest to use formula editor instead of figure, especially considering the final quality. Figure 6: improve the quality.

I strongly encourage the Authors to correct the paper, as in my opinion it presents very interesting study and, after improvements mentioned above, might constitute a valuable paper.

Author Response

PLEASE SEE THE ATTACHEMENT

Reviewer 3 Report

The paper is well structured and organized, presenting in a clear way the context and background of the research, as well as the main goal of the research and the goals of the paper. Methodologically, the paper describes and justifies why the research methods applied were chosen and the authors are aware of the limitations the chosen methodology implies. The expected results are also evident and are well systemized. However, the correlation between the different issues raised within the research are, sometimes, not clearly established and, above all, the paper doesn’t explore the importance the relevance of the co-operationalization of the aspects and fields needed to assure local engagement towards co-creation based on e-participation, e-planning and e-governance – structuring the co-design of public policies on a municipal government level. It is not clear how to effectively turn operative the mentioned guidelines of City Statute for and with community intelligence – constituting the common ground for smart and sustainable cities. Finally, the morphological framework could be better explored when addressing issues regarding the intelligence of cities.

Reviewer 4 Report

The article raises very important issues from the point of view of urban development in accordance with the concept of a smart city.

However, the way of scientific eduction should take into account the following elements:

The authors didn't mention neither in the subject nor in the abstract or introduction, that their article was limited to Brazilian cities only.

The abstract is not fully transparent. Is it a real scientific gap or perhaps it should be a bit redefined? Moreover:

there's no explanation why the four areas were chosen, a reader doesn't recognize the mentioned five guidelines.

The introduction would benefit from arranging the paragraphs in order of logical thread.

It would be worth starting by outlining the topic in general, and then clearly indicating that the article is based on and applies to the law in force in Brazil, and the analysis and subsequent recommendations will be narrowed down to Brazilian cities.

It is also worth referring to the urban situation in the world, defining the concept of an intelligent city and providing how the authors understand terms mentioned in the article, basing on the global literature.

City Statute Guidelines - it has not been mentioned previously that the City Statute Guidelines document concerns Brazilian law only - the readers could feel confused.

There's no summary or authors' comments for the Table 1 provided.

The methodology of Bibliographic research should be described more in details. It is currently unclear how the exact keywords were chosen. How did the search take place? It is worth supplementing with information about the AND, OR connectors used. The methodological part would benefit if it were supplemented with a graph depicting the entire search process.

Were there posed any research questions that authors try to answer to? If yes, please, make them visible to the readers.

In Figure 1, arrows to fields containing information about how many publications have been rejected should lead from the fields: 2027 records (and information about rejected should appear here) and 245 screening records (also here a link, how many from this group of publications , the authors gave up)

row 202 - why Humanities appears here? Please, refer to the previous paragraph

Results and discussion - (Fig.3) does the share of each area mean anything to the formulated guidelines?

Row 299 Has Guedes et al been mentioned before in the text? Why does his work appear in this place?

Please review the discussion, it will benefit from putting thoughts (paragraphs) in logical order. It is important that one paragraph results directly from the other one.

How the experts were chosen, on what basis?

Round 2

Reviewer 1 Report

The authors have significantly improved the quality of the article. It can be published in its present form.

Reviewer 2 Report

The paper has been corrected according to my previous comments and in my opinion it can be published in the current form.

Reviewer 4 Report

Thank you for the introduction of changes.